# Ethiopian antimicrobial consumption trends in human health sector: A surveillance report 2020–2022

Hailemariam Eshete[1], Melaku Tileku[2], Abiyot Aschenaki[1], Eshetu Shiferaw[3], Haregewoin Mulugeta[1], Mengistab Teferi[4], Teshita Shute[1], Asnakech Alemu[1], Heran Gerba[5], Atalay Mulu Fentie[2]*

1 Pharmacovigilance and Clinical trial lead executive office, Ethiopian Food and Drug Authority, Addis Ababa, Ethiopia, 2 School of Pharmacy, College of Health Sciences, Addis Ababa University, Addis Ababa, Ethiopia, 3 Department of Pharmacy, Yekatit 12 Hospital Medical College, Addis Ababa, Ethiopia, 4 Essential Drugs and Medicines, World Health Organization Ethiopia Country Office, Addis Ababa, Ethiopia, 5 Ethiopian Food and Drug Authority, Addis Ababa, Ethiopia

* atalay.mulu@aau.edu.et

## Abstract

### Background

Antimicrobial resistance (AMR) poses a severe global health threat, driven by the overuse and misuse of antimicrobials across the human, agricultural, and veterinary sectors. To combat this, global and national AMR prevention and containment strategies have been implemented, necessitating continuous monitoring of antimicrobial consumption (AMC) as an integral part of antimicrobial stewardship interventions.

### Objective

This study aims to assess and analyze trends in AMC in Ethiopia from 2020 to 2022, with the goal of informing national and sub-national strategies to combat AMR.

### Methods

A three-year AMC surveillance was conducted from 2020 to 2022. Data on locally manufactured and imported antimicrobials were collected from local manufacturers and Ethiopian Food and Drug Authority (EFDA)-regulated ports of entry. AMC was analyzed using the WHO GLASS AMC tool, with antimicrobials categorized using the WHO Anatomical Therapeutic Chemical (ATC) classification system. Consumption was measured in Defined Daily Doses (DDDs) and DDD per 1,000 inhabitants per day (DID), normalized using population estimates from the World Population Prospects for Ethiopia.

### Results

The total AMC in Ethiopia increased from 432 million DDDs in 2020 to 485 million DDDs in 2022. The DID rose from 10.63 in 2020 to 11.34 in 2022. Antibacterials dominated consumption, comprising 98.87% in 2020, 95.96% in 2021, and 99.79% in 2022. Penicillins (J01C)

**Data availability statement:** All the data is available in the manuscript.

**Funding:** The author(s) received no specific funding for this work.

**Competing interests:** The authors have declared that no competing interests exist.

**Abbreviation:** AMC, Antimicrobial Consumption; AMR, Antimicrobial Resistance; ATC, Anatomical Therapeutic Classification; AWaRe, Access, Watch, and Reserve; COVID-19, Coronavirus Disease 2019; DDD, Defined Daily Dose; DID, DDD per 1,000 inhabitants per day; EFDA, Ethiopian Food and Drug Authority; GLASS, Global Antimicrobial Resistance and Use Surveillance System; RoAs, Route of Administrations; WHO, World Health Organization.

and quinolones (J01M) were the most consumed antimicrobials. As per the Ethiopian AWaRe classification, the majority of antibacterial agents consumed were in the Access group, accounting for 71.14% in 2020, 70.65% in 2021, and 74.2% in 2022. Oral formulations consistently made up over 87% of the total consumption each year. Reliance on imported antimicrobials remained high, with imports comprising 64.76% in 2020 and 74.47% in 2022.

## Conclusion

The increasing trend in AMC in Ethiopia from 2020 to 2022 underscores the urgent need to establish and strengthen national, sub-national, and facility-level surveillance and reporting systems to better monitor and ensure rational antimicrobial use.

## Introduction

Antimicrobial resistance (AMR) is a critical global health issue of the 21$^{st}$ century [1], driven by the overuse and misuse of antimicrobials in human and animal health, and as well as in agriculture. This widespread misuse accelerates the development of resistant microorganisms [2], leading to ineffective treatment, prolonged illness, increased mortality, and higher healthcare costs [3–5]. Ethiopia, like many other low- and middle-income countries, faces significant challenges in combating AMR due to limited healthcare infrastructure, inadequate surveillance systems, and insufficient public awareness [6–12].

Monitoring antimicrobial consumption (AMC) is essential for generating evidence for policy and practice in combating AMR effectively [13–16]. Surveillance of AMC serves as a proxy indicator to identify misuse and overuse, which are the primary drivers of resistance [15,16] and AMC data provide critical insights into prescribing practices, trends in consumption, and areas requiring targeted interventions [17–19]. Such data are vital for formulating policies and programs aimed at promoting the rational use of antimicrobials and mitigating resistance. However, comprehensive data on national AMC trends over extended periods have been scarce in Ethiopia, which hampers the development of targeted interventions and policies for rational antimicrobial use [20]. Previous studies have highlighted issues such as self-medication, over-the-counter availability of antimicrobials, and non-adherence to treatment guidelines [21], but these studies did not aim to reveal the complete AMC all over the country [22,23]. This knowledge gap hinders the development of effective AMR containment strategies, underscoring the urgent need for a thorough analysis of antimicrobial use trends [24–28].

In Ethiopia, most antimicrobials are imported from foreign manufacturers or suppliers, with a few sourced from local manufacturers. The public sector primarily supplied by the Ethiopian Pharmaceutical Supply Services, while private healthcare facilities including community pharmacies obtain antimicrobials from private importers and wholesalers. This study aims to address this gap by assessing and analyzing AMC trends in Ethiopia exclusively in the human health sector from 2020 to 2022 with respect to overall consumption across the years, and different class of medications consumption patterns. The findings will offer a detailed overview of AMC patterns at the national level, providing essential insights for policymakers, healthcare providers, and other stakeholders to make evidence-based decisions aimed at curbing AMR in Ethiopia.

## Methods

### Study area and design

The study encompassed the entire country of Ethiopia, with specific data collection points at Ethiopian Food and Drug Authority (EFDA) regulated all ports of entry (Cargo Terminal

and Kality in Addis Ababa, Modjo, Moyale, Semera, and Dire Dawa). Data were also collected from local manufacturers to ensure comprehensive coverage across the country and analysed AMC in Ethiopia over a three-year period from 2020 to 2022.

## Data source and validation

Data were sourced from the EFDA electronic regulatory information system (eRIS) import records that included only the human medicines and local manufacturers producing antimicrobials for human use in Ethiopia only. This provides comprehensive information on the importation and production of antimicrobial products for human use. The collected data were rigorously validated by cross-referencing with importers' records and local manufacturing data where possible to ensure accuracy and reliability. This validation process was crucial for obtaining a precise understanding of the total volume of antimicrobials available in the country.

## Eligibility criteria

Antimicrobials, including antibacterials, antivirals, antimycotics, and antifungals, were included in the surveillance if they were classified under the WHO Anatomical Therapeutic Chemical (ATC) Classification system and listed in the ATC/DDD 2023 index, and had an associated Defined Daily Dose (DDD). Antimicrobials that lacks ATC code or DDD not assigned in the WHO ATC system were excluded from the analysis (S1 file).

## Classification

Antimicrobials are classified according to the WHO ATC classification system [29]. The classifications include five main antimicrobial classes and presented below with their respective ATC code:

- Antibacterials (J01, A07AA, P01AB)
- Antimycotics for systemic use (J02) and antifungals for systemic use (D01B)
- Antivirals for systemic use (J05)
- Drugs for the treatment of tuberculosis (J04A)
- Antimalarials (P01B)

## Data collection instruments and procedures

AMC data were systematically collected using data extraction sheets from eRIS and manufacturing records. The WHO GLASS AMC data entry tool was utilized for data entry and analysis to maintain alignment with global standards. Data on AMC were collected annually from 2020 to 2022. For each year, data collection was conducted from January to December, ensuring a comprehensive annual dataset. The data were compiled and analyzed early in the following year to allow for a timely assessment of trends and patterns. This timeline was chosen to capture seasonal variations and other temporal factors that could influence consumption patterns (Fig 1).

## Data quality management

To ensure data quality, several measures were implemented. Training sessions were provided for data collectors regarding the data collection tools, procedures, and data abstraction sheets. The WHO GLASS AMC Macro Automated Validation Tool was employed to ensure data

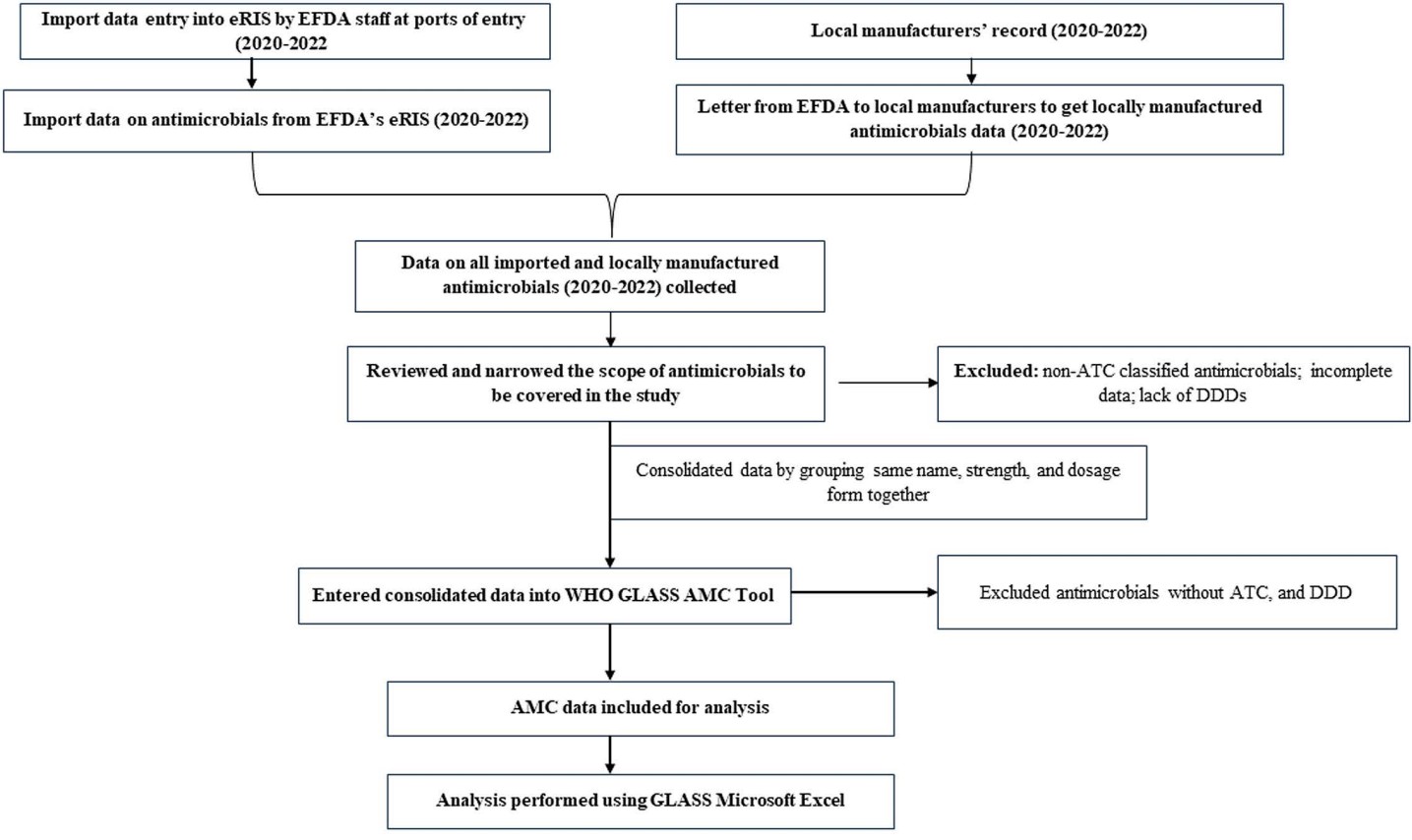

**Fig 1. Data collection and cleaning methodology for the Ethiopia national antimicrobial consumption analysis.**

completeness, consistency, accuracy, and conformity. Additionally, data verification involved crosschecking import data with records from importers to ensure accuracy and consistency throughout the process.

## Metrics and data analysis

The volume of antimicrobials consumed was presented using two metrics: DDD and the weight of antimicrobial substances in metric tons. Consumption was reported for total care, incorporating both imported and locally manufactured data. It was expressed as DDDs per 1,000 inhabitants per day (DID), and relative consumption was also calculated. Population data were obtained from the World Population Prospects 2020–2022 and adjusted according to the system coverage level provided by the country, which represent about 95% national consumption (Table 1). To assess the relative consumption of antibacterials by AWaRe categories, both the WHO 2023 (https://www.who.int/medicines/areas/rational_use/awareness/en/) and Ethiopia 2020 AWaRe classifications (http://www.efda.gov.et/publication/ethiopian-essential-medicines-list-sixth-edition-2020/?lang=amh) were applied.

Data analysis was conducted using the WHO GLASS AMC Microsoft Excel-Based Tool. The collected data were cleaned, validated, and organized for analysis. Descriptive statistics were used to summarize the data, and trends in AMC were analyzed using DDDs per substance, total volume in DDDs, metric tonnes, and DID. Results were presented in both tabular and graphical formats to facilitate easy interpretation.

**Table 1. Data contextual information as per GLASS database.**

| Year | Population (WPP*) | Population Coverage | Population Used for Analysis | Level | Sources of Data | ATC Classes Reported** |
|------|-------------------|---------------------|------------------------------|-------|-----------------|-------------------------|
| 2022 | 123,379,924 | 95% | 117,210,927 | Import/Local Manufacturing | Production for domestic market, Import records | J01, J02, J05, P01AB, P01B |
| 2021 | 120,283,026 | 95% | 114,268,874 | Import/Local Manufacturing | Production for domestic market, Import records | D01BA, J01, J02, J05, P01AB, P01B |
| 2020 | 117,190,911 | 95% | 111,331,365 | Import/Local Manufacturing | Production for domestic market, Import records | D01BA, J01, J02, J05, P01AB, P01B |

*WPP: estimates by the United Nations, adjusted for Ethiopia's population during the study period.

**D01BA (antifungals for systemic use) was stockout in the year 2022 and not included in the AMC report. Throughout the reporting period (2020–2022), A07AA (Intestinal antibiotics) were not consumed and not included in the AMC report.

### Ethical considerations

A permission letter from the EFDA General Director was submitted to local manufacturers, EFDA's central branch office (responsible for controlling ports of entry), and the Medicines Registration and Licensing Directorate to obtain all necessary data. All data were handled with strict confidentiality, and measures were taken to ensure that individual data sources were anonymized and protected throughout the analysis process.

## Results

### Overall antimicrobial consumption in tons and DDD

The total volume of antimicrobials consumed, regardless of group, was 532.23 tons and 431,972,986.1 DDD in 2020, 692.975 tons and 547,659,483.1 DDD in 2021, and 608.72 tons and 485,320,971.8 DDD in 2022. The three-year average total AMC in Ethiopia was 488,317,813.7 DDD (Table 2). The majority of the consumed antimicrobials were from the antibacterial (J01, A07AA, P01AB) group. A notable increase in consumption was observed in 2021 (547.7 million DDD), followed by a decrease in 2022 (488.3 million DDD). The antimalarial consumption was also significantly decreased in 2022 to 169.6 thousand DDD as compared to 5.68 million DDD in 2021 (Table 2).

### Overall antimicrobial consumption in DID

The AMC in Ethiopia measured in DID was 10.630 in 2020, 13.131 in 2021 and 11.344 in 2022. Almost all antimicrobials consumed during this period were from the antibacterial group (J01, A07AA, P01AB), accounting for 98.87% in 2020, 95.96% in 2021, and 99.79% in 2022. Antimycotics and antifungals for systemic use (J02, D01B) group accounted for 0.94%, 2.59%, and 0.09%, respectively for the year 2020, 2021 and 2022. The orally administered antimicrobial consumption was accounted 87.96%, 87.5% and 91.24% in 2020, 2021 and 2022, respectively. The highest consumption of parenteral antimicrobials were observed in 2021, accounting for 12.5% of the total AMC. The total DID and respective antimicrobial group DID for each year are summarized in Table 3.

### Antimicrobial consumption by source

In 2020, all consumed antivirals for systemic use and antimycotics/antifungals for systemic use were from import, whereas in 2021, 1.64% of antivirals for systemic use and 51.93% of antimycotics/antifungals for systemic use were from local manufacturers. By 2022, the proportion of locally manufactured antivirals for systemic use increased to 10%, while no

**Table 2. A three-year antimicrobial consumption expressed in tons and DDD in Ethiopia.**

| Pharmacological subgroups | 2020 | | 2021 | | 2022 | | Three years DDD Average |
|---|---|---|---|---|---|---|---|
| | Tons | DDD | Tons | DDD | Tons | DDD | |
| Antibacterial (JO1, A07AA, P01AB) | 531.1 | 427,069,849.1 | 687.8 | 525,404,773.7 | 608.1 | 484,188,917.8 | 478,887,846.9 |
| Antimycotics and antifungals for systemic use (JO2, D01B) | 0.87 | 4,264,382.4 | 3.28 | 1,4036,306.0 | 0.10 | 500,000.0 | 6,266,896.13 |
| Antimalarials (P01B) | 0.14 | 608,857.1 | 1.59 | 5,676,618.4 | 0.05 | 169,607.1 | 2,151,694.2 |
| Antivirals for systemic use (JO5) | 0.12 | 29,897.5 | 0.31 | 2,541,785.0 | 0.47 | 462,446.9 | 1,011,376.47 |
| **Total** | **532.23** | **431,972,986.1** | **692.98** | **547,659,483.1** | **608.72** | **485,320,971.8** | **488,317,814** |

*DDD: Defined Daily Dose*

**Table 3. A three years antimicrobial consumption in Ethiopia as per DDD per 1000 inhabitants per day (DID).**

| Category | | 2020 | | 2021 | | 2022 | | A 3-year average | |
|---|---|---|---|---|---|---|---|---|---|
| | | DID | % | DID | % | DID | % | DID | % |
| **Antimicrobial group** | Antibacterials* | 10.51 | 98.87 | 12.60 | 95.96 | 11.32 | 99.79 | 11.48 | 98.10 |
| | Antimycotics and antifungals for systemic use | 0.10 | 0.94 | 0.34 | 2.59 | 0.010 | 0.09 | 0.15 | 1.30 |
| | Antimalarials | 0.015 | 0.14 | 0.136 | 1.04 | 0.004 | 0.04 | 0.024 | 0.44 |
| | Antiviral for systemic use | 0.001 | 0.01 | 0.061 | 0.46 | 0.011 | 0.10 | 0.0467 | 0.21 |
| | **Total** | **10.63** | | **13.13** | | **11.34** | | **11.70** | |
| **ROA** | Oral | 9.35 | 87.96 | 11.49 | 87.5 | 10.35 | 91.24 | 10.397 | 88.88 |
| | Parenteral | 1.28 | 12.04 | 1.64 | 12.5 | 1.00 | 8.76 | 1.301 | 11.12 |
| | **Total** | **10.63** | | **13.13** | | **11.34** | | **11.70** | |

*Agents against amoebiasis and other protozoal diseases such as metronidazole and tinidazole are included as antibacterials. DID: Defined Daily Dose per 1000 inhabitants per day; ROA: Route of Administration.*

**Table 4. Antimicrobial consumption in Ethiopia as per the source of antimicrobials and DDD/1000 inhabitants/day (DID) per antimicrobial source.**

| Antimicrobial group | 2020, DID (%) | | | 2021, DID (%) | | | 2022, DID (%) | | |
|---|---|---|---|---|---|---|---|---|---|
| | Local* | Imported | Total | Local* | Imported | Total | Local* | Imported | Total |
| Antibacterials | 3.746 (35.65) | 6.763 (64.35) | **10.509** | 3.090 (24.53) | 9.508 (75.47) | **12.598** | 2.895 (25.58) | 8.423 (74.42) | **11.318** |
| Antiviral for systemic use | 0.00 (0.00) | 0.001 (100.0) | **0.001** | 0.001 (1.64) | 0.060 (98.34) | **0.061** | 0.001 (10.00) | 0.009 (90.00) | **0.01** |
| Antimycotics and antifungals for systemic use | 0.00 (0.00) | 0.105 (100.0) | **0.105** | 0.175 (51.93) | 0.162 (48.07) | **0.337** | 0.00 (0.00) | 0.012 (100.00) | **0.012** |
| Antimalarials | 0.00 (0.00) | 0.015 (100.0) | **0.015** | 0.00 (0.00) | 0.136 (100.0) | **0.136** | 0.00 (0.00) | 0.004 (100.00) | **0.004** |
| Total | **3.746 (35.24)** | **6.884(64.76)** | 10.63 | **3.266(24.87)** | **9.866(75.13)** | 13.131 | **2.896(25.53)** | **8.448(74.47)** | 11.344 |

*Locally manufactured. DID: Defined Daily Dose per 1000 inhabitants per day.*

antimycotics or antifungals were locally manufactured. Throughout the entire reporting period (2020–2022), all consumed antimalarial were imported (Table 4).

As described in Fig 2 below, most of the antimicrobials consumed in Ethiopia were imported. In 2020, 64.76% of antimicrobials consumed (measured in DID) were imported. This reliance on imports increased to 75.13% of antimicrobials in 2021 and 74.47% in 2022.

## Antibacterial consumption by subgroups

Penicillin's (J01C) were the most commonly consumed antibacterial subgroup during the reporting period, accounting for 39.65% in 2020, 35.88% in 2021, and 37.78% in 2022.

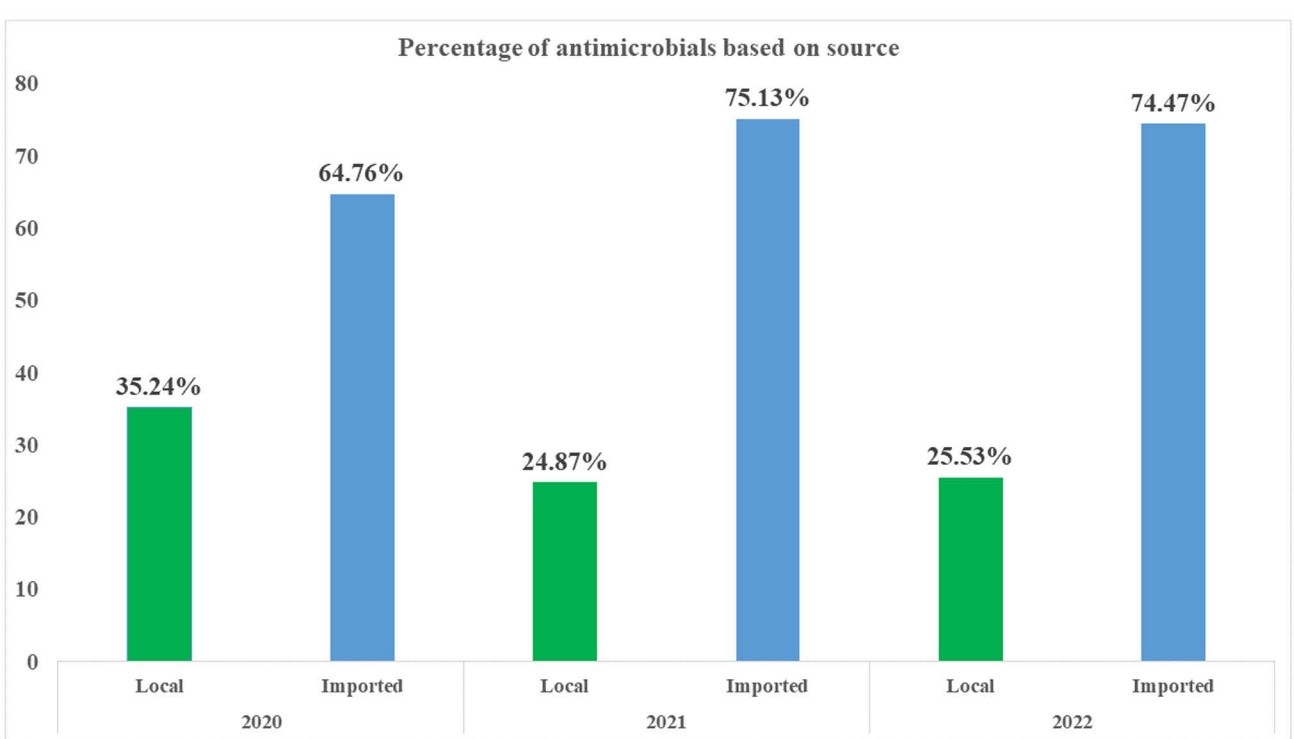

**Fig 2. Antimicrobial consumption in Ethiopia based on the source.**

Quinolones (J01M) followed, with consumption levels of 25.59% in 2020, 20.56% in 2021, and 21.80% in 2022. Agents against amoebiasis and other protozoal diseases (P01AB) were also significant, contributing 15.0% in 2020, 13.5% in 2021, and 15.1% in 2022. The consumption trend of extended-spectrum penicillin's (J01CA) increased significantly from 66.9% in 2020 to 79.0% in 2022. Conversely, the use of macrolides and lincosamides (J01F) decreased from 6.22% in 2020 to 1.34% in 2022. There was also a decline in the consumption of 1st and 2nd generation cephalosporin, while the use of 3rd generation cephalosporin's(J01DD) increased over the same period. Additionally, the consumption of tetracycline's (J01A) rose from 6.17% in 2020 to 13.02% in 2022 (Table 5).

## Antibacterial consumption by AWaRe classification

The Access category, which includes antimicrobials recommended as the first or second choice for most common infections, dominated consumption in Ethiopia, accounting for 71.14% in 2020, 70.65% in 2021, and 74.20% in 2022. This was slightly higher compared to the WHO AWaRe classification, where Access category consumption was 66.20% in 2020, 67.60% in 2021, and 72.30% in 2022. The Watch category, comprising antimicrobials with a higher potential for resistance, made up 24.64% of consumption in 2020, 24.69% in 2021, and 24.73% in 2022 in Ethiopia. The Reserve category, which includes last-resort antimicrobials, had minimal usage, with 0.03% in 2020, 0.06% in 2021, and 0.02% in 2022 (Table 6).

## DU75 and DU90 of antimicrobial consumption

The antimicrobials that constituted 75% (DU75) of the overall consumption from 2020 to 2022 included amoxicillin, ciprofloxacin, metronidazole, procaine benzyl penicillin, and

**Table 5. Antimicrobial consumption in Ethiopia based on antibacterial subgroup.**

| Antibacterial subgroup | | 2020 | | 2021 | | 2022 | |
|---|---|---|---|---|---|---|---|
| | | DID | % | DID | % | DID | % |
| Penicillin (J01C) | | 4.17 | 39.65 | 4.52 | 35.88 | 4.28 | 37.78 |
| ATC Level 4 | Penicillin with extended spectrum (J01CA) | 2.79 | 66.9 | 3.30 | 73.0 | 3.38 | 79.0 |
| | Beta-lactamase sensitive penicillins (J01CE) | 1.07 | 25.6 | 0.82 | 18.2 | 0.67 | 15.6 |
| | Beta-lactamase resistant penicillins (J01CF) | 0.31 | 7.5 | 0.40 | 8.8 | 0.23 | 5.4 |
| | Combinations of penicillins, incl. beta-lactamase inhibitors (J01CR) | 0.00 | 0.00 | 0.00 | 0.00 | 0.00 | 0.00 |
| Other beta-lactam antibacterials (J01D) | | 0.41 | 3.92 | 1.55 | 12.28 | 0.68 | 6.05 |
| ATC Level 4 | 1st generation cephalosporin's (J01DB) | 0.20 | 49.3 | 0.56 | 36.4 | 0.17 | 25.5 |
| | 2nd generation cephalosporin's (J01DC) | 0.02 | 5.6 | 0.01 | 0.86 | 0.002 | 0.07 |
| | 3rd generation cephalosporin's (J01DD) | 0.18 | 44.8 | 0.97 | 62.5 | 0.51 | 74.4 |
| | 4th generation cephalosporin's (J01DE) | 0.004 | 0.20 | 0.001 | 0.09 | 0.003 | 0.05 |
| | Monobactams (J01DF) | 0.00 | 0.00 | 0.00 | 0.00 | 0.00 | 0.00 |
| | Other cephalosporins & penems (J01DI) | 0.00 | 0.00 | 0.00 | 0.00 | 0.00 | 0.00 |
| | Carbapenems (J01DH) | 0.03 | 0.13 | 0.04 | 0.19 | 0.002 | 0.04 |
| Quinolone (J01M) | | 2.69 | 25.59 | 2.59 | 20.56 | 2.47 | 21.80 |
| Anti-amoebiasis and other protozoal diseases (P01AB) | | 1.58 | 15.00 | 1.70 | 13.50 | 1.709 | 15.10 |
| Macrolides, and lincosamides (J01F) | | 0.65 | 6.22 | 0.51 | 4.02 | 0.15 | 1.34 |
| Tetracycline's (J01A) | | 0.65 | 6.17 | 1.01 | 7.99 | 1.47 | 13.02 |
| Sulfonamides and trimethoprim (J01E) | | 0.29 | 2.77 | 0.62 | 4.96 | 0.49 | 4.34 |
| Aminoglycoside (J01G) | | 0.05 | 0.44 | 0.07 | 0.56 | 0.05 | 0.46 |
| Amphenicols (J01B) | | 0.02 | 0.17 | 0.03 | 0.24 | 0.01 | 0.09 |
| Other antibacterials (J01X) | | 0.01 | 0.07 | 0.00 | 0.00 | 0.00 | 0.02 |

*DID: Defined Daily Dose per 1000 inhabitants per day*

**Table 6. Antimicrobial consumption in Ethiopia based on AWaRe classification.**

| AWaRe classification | | 2020 | | 2021 | | 2022 | |
|---|---|---|---|---|---|---|---|
| | | DID | % | DID | % | DID | % |
| Ethiopia AWaRe category | | | | | | | |
| | Access | 7.56 | 71.14 | 9.28 | 70.65 | 8.42 | 74.20 |
| | Watch | 2.62 | 24.64 | 3.24 | 24.69 | 2.81 | 24.73 |
| | Reserve | 0.003 | 0.03 | 0.01 | 0.06 | 0.003 | 0.02 |
| | Not classified | 0.45 | 4.19 | 0.60 | 4.60 | 0.12 | 1.05 |
| | Total | **10.63** | | **13.13** | | **11.34** | |
| WHO AWaRe category | | | | | | | |
| | Access | 6.95 | 66.20 | 8.51 | 67.60 | 8.19 | 72.30 |
| | Watch | 3.55 | 33.80 | 4.09 | 32.40 | 3.13 | 27.7 |
| | Reserve | 0.00 | 0.00 | 0.00 | 0.00 | 0.00 | 0.00 |
| | Not classified | 0.00 | 0.00 | 0.00 | 0.00 | 0.00 | 0.00 |
| | Total | **10.51** | | **12.60** | | **11.32** | |

*DID: Defined Daily Dose per 1000 inhabitants per day*

doxycycline. These antimicrobials also prominently featured in the 90% consumption (DU90) category, along with additional antimicrobials such as azithromycin, cloxacillin, trimethoprim/sulphamethoxazole, and ceftriaxone.

Amoxicillin, ciprofloxacin, and metronidazole were the most commonly consumed oral formulations within the DU75 category, consistently across the years. Procaine benzyl penicillin was the most consumed parenteral antimicrobial, forming a substantial portion of the DU75 category each year. Ceftriaxone followed as the next most consumed parenteral antimicrobial, particularly in the DU90 category, especially during 2021 and 2022 (Table 7).

## Antimalarial consumption based on substance

As shown in Fig 3, antimalarial consumption was highest in 2021, with artemether plus lumefantrine accounting for the largest share (0.127 DID out of 0.136 DID). Notably, artemether consumption dropped to zero in 2021, compared to 0.004 DID out of 0.015 DID in 2020. Additionally, artesunate consumption in 2021 was 0.009 DID, but it was zero in both 2020 and 2022.

**Table 7. DU75 and DU90 antimicrobial consumption in Ethiopia as per route of administration.**

| Antimicrobial substance | | 2020 | | | 2021 | | | 2022 | | |
|---|---|---|---|---|---|---|---|---|---|---|
| | | DID | DU75 | DU90 | DID | DU75 | DU90 | DID | DU75 | DU90 |
| Overall | Amoxicillin (J01CA04) | 2.63 | 24.73 | 24.73 | 2.91 | 22.15 | 22.15 | 3.06 | 27.00 | 27.00 |
| | Ciprofloxacin (J01MA02) | 2.42 | 22.74 | 22.74 | 2.26 | 17.22 | 17.22 | 2.29 | 20.24 | 20.24 |
| | Metronidazole (J01XD01) | 1.44 | 13.58 | 13.58 | 1.66 | 12.65 | 12.65 | 1.62 | 14.35 | 14.35 |
| | Procaine benzyl penicillin (J01CE09) | 1.06 | 10.01 | 10.01 | 0.81 | 6.19 | 6.19 | 0.66 | -- | 5.87 |
| | Doxycycline (J01AA02) | 0.65 | 6.10 | 6.10 | 1.00 | 7.67 | 7.67 | 1.47 | 12.99 | 12.99 |
| | Azithromycin (J01FA10) | 0.34 | -- | 3.22 | 0.40 | -- | 3.08 | 0.15 | -- | -- |
| | Cloxacillin (J01CF03) | 0.31 | -- | 2.93 | 0.39 | -- | 3.03 | 0.23 | -- | 2.05 |
| | Trimethoprim/ sulphamethoxazole (J01EE01) | 0.29 | -- | 2.74 | 0.62 | 4.76 | 4.76 | 0.49 | -- | 4.33 |
| | Ceftriaxone (J01DD04) | 0.13 | -- | -- | 0.63 | 4.59 | 4.59 | 0.21 | -- | 1.91 |
| | Norfloxacin (J01MA06) | 0.27 | -- | 2.55 | 0.33 | -- | -- | 0.17 | -- | -- |
| | Cephalexin (J01DB01) | 0.20 | -- | 1.89 | 0.56 | -- | 4.28 | 0.17 | -- | -- |
| | Ampicillin (J01CA01) | 0.16 | -- | -- | 0.39 | -- | 2.97 | 0.31 | -- | 2.78 |
| | Cefixime (J01DD08) | 0.04 | -- | -- | 0.36 | -- | 2.73 | 0.28 | -- | 2.49 |
| | **Number of antimicrobials** | | **5** | **10** | | **7** | **12** | | **4** | **10** |
| Oral | Amoxicillin (J01CA04) | 2.63 | 28.13 | 28.13 | 2.91 | 25.32 | 25.32 | 3.06 | 29.60 | 29.60 |
| | Ciprofloxacin (J01MA02) | 2.39 | 25.60 | 25.60 | 2.18 | 18.93 | 18.93 | 2.29 | 22.19 | 22.19 |
| | Metronidazole (J01XD01) | 1.44 | 15.39 | 15.39 | 1.63 | 14.23 | 14.23 | 1.62 | 15.66 | 15.66 |
| | Doxycycline (J01AA02) | 0.65 | 6.94 | 6.94 | 1.00 | 8.76 | 8.76 | 1.47 | -- | 14.24 |
| | Azithromycin (J01FA10) | 0.34 | -- | 3.66 | 0.40 | -- | 3.52 | 0.15 | -- | -- |
| | Cloxacillin (J01CF03) | 0.31 | -- | 3.34 | 0.39 | -- | 3.46 | 0.23 | -- | -- |
| | Trimethoprim/sulphamethoxazole (J01EE01 | 0.29 | -- | 3.11 | 0.62 | 5.43 | 5.43 | 0.49 | -- | 4.75 |
| | Norfloxacin (J01MA06) | 0.27 | -- | 2.90 | 0.33 | -- | -- | 0.17 | -- | -- |
| | Cephalexin (J01DB01) | 0.20 | -- | 2.10 | 0.56 | 4.89 | 4.89 | 0.17 | -- | -- |
| | Ampicillin (J01CA01) | 0.16 | -- | -- | 0.37 | -- | 3.24 | 0.27 | -- | -- |
| | Cefixime (J01DD08) | 0.04 | -- | -- | 0.36 | -- | 3.11 | 0.28 | -- | -- |
| | **Number of antimicrobials** | | **4** | **9** | | **6** | **10** | | **3** | **5** |
| Parenteral | Procaine benzyl penicillin (J01CE09) | 1.06 | 82.81 | 82.81 | 0.81 | 49.59 | 49.59 | 0.67 | 66.81 | 66.81 |
| | Ceftriaxone (J01DD04) | 0.13 | -- | 10.30 | 0.60 | 36.76 | 36.76 | 0.22 | 21.71 | 21.71 |
| | Gentamycin (J01GB03) | 0.05 | -- | -- | 0.07 | -- | -- | 0.05 | -- | 5.26 |
| | Ciprofloxacin (J01MA02) | 0.03 | -- | -- | 0.09 | -- | 5.21 | 0.0004 | -- | -- |
| | **Number of antimicrobials** | | **1** | **2** | | **2** | **3** | | **2** | **3** |

*DID: Defined Daily Dose per 1000 inhabitants per day, DU75: 75% drug utilization index; DU90: 90% drug utilization index*

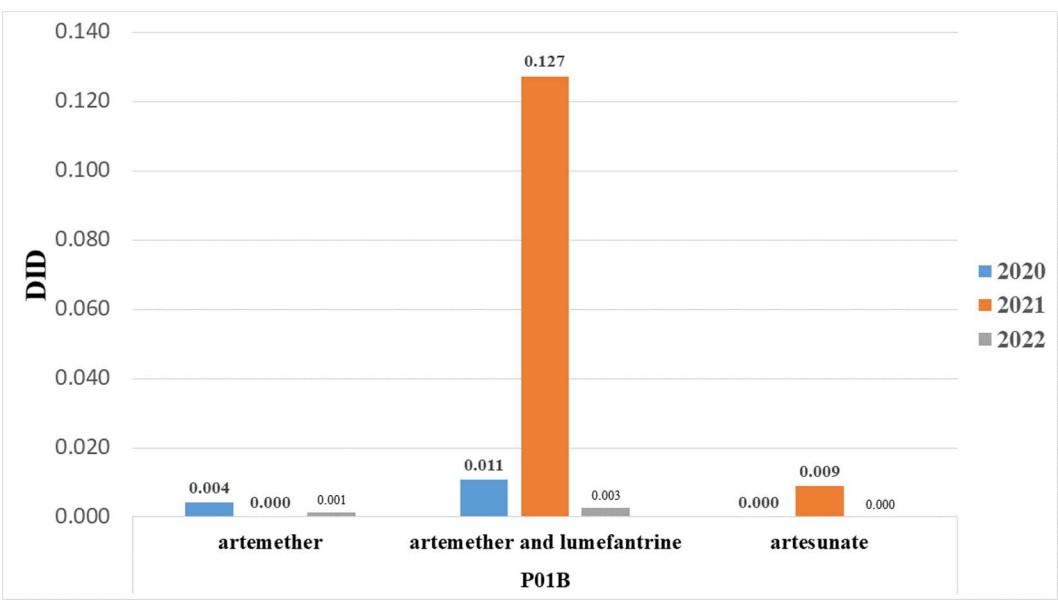

**Fig 3. Antimalarial consumption by substance per DDD per 1000 inhabitants per day (DID).**

## Discussion

The analysis of AMC in Ethiopia from 2020 to 2022 reveals significant trends and insights crucial for addressing the challenges posed by AMR in the country. The findings highlight patterns in overall consumption, sources of antimicrobials, and the distribution across different antibacterial subgroups, routes of administration, and AWaRe classification.

The AMC in Ethiopia showed significant increases in 2021, followed by declines in 2022. The surge in 2021 might be attributed to the lifting of social restrictions or lockdown and other infection prevention measures, which led to an increase in non-COVID infections and antimicrobial use. Additionally, it might be due to shift in imports from COVID-related products to other health commodities like antimicrobials. The heightened demand during the second wave of COVID-19 (2021) also contributed, as antimicrobials were often used empirically to manage suspected bacterial infections. The AMC from 2020 to 2022 in Ethiopia, also revealed a notable increase from the previous three years (2017–2019), with values rising from 8.50 DID in 2017 to 10.00 DID in 2019 and further to an average of 11.70 DID in the 2020–2022 period, peaking at 13.13 DID in 2021[20]. This increase may be attributed to a rise in antimicrobial use during the different stages of the COVID-19 pandemic, as compared to the pre-pandemic period. Additionally, the 2017–2019 data did not fully accounted for incomplete records, which could also have influenced the analysis and contributed to the observed trends.

The AMC in Sri Lanka in 2017 was 16.26 DID which slightly higher that the Ethiopian average three years AMC [30]. In contrast, a significantly higher AMC was observed in Uganda and Tanzania during the same periods. Uganda, for instance, experienced a peak DID of 123.30 in 2019, followed by a significant decline to 29.02 in 2021, reflecting a possible impact of the COVID-19 pandemic on healthcare practices and antimicrobial use [31]. Similarly, in Tanzania the AMC in DID before the pandemic was93.11 in 2017 and 80.80 in 2019, indicating a consistently high level of AMC [32]. In contrast, high-income countries like the United Kingdom and the United States either maintained stable or declining DID values during the pandemic, reflecting the effectiveness

of established antimicrobial stewardship programs in these regions [33]. The comparison underscores the variability in AMC across different regions; with Ethiopia, showing a more controlled but increasing trend, while Uganda and Tanzania exhibited higher but more fluctuating patterns, and high-income countries maintained more stable consumption levels.

The data indicated Ethiopia's strong reliance on imported antimicrobials, with over 64% of antimicrobials being imported each year. This dependency on imports underscores the challenges faced by the local pharmaceutical manufacturing sector in meeting the national demand for antimicrobials. The consistently high percentage of imported antimicrobials, peaking at 75.13% in 2021 as compared to the 2020 (during COVID-19 related lockdown), reflects the need for policies to strengthen local production capabilities. Similarly, other studies in African countries such as Nigeria and Kenya have shown that local production capabilities are limited [34]. This reliance poses significant risks during global supply chain disruptions, as seen during the COVID-19 pandemic, highlighting vulnerabilities in accessing essential medicines. The pandemic-induced disruptions in global supply chains have underscored the urgent need to strengthen the Ethiopian local pharmaceutical manufacturing, which would enhance healthcare system resilience and reduce dependency on international supply chains like the situation happened during COVID-19 [35,36].

Penicillins and quinolones were the most commonly consumed antibacterial subgroups throughout the reporting period, accounting for significant proportions of the total AMC. Penicillins comprised 39.65% in 2020, 35.88% in 2021, and 37.78% in 2022, while quinolones made up 25.59%, 20.56%, and 21.80%, respectively. Similarly, penicillin's were the most commonly consumed antimicrobials in Uganda, Tanzania and Sri Lanka. However, the second most consumed antimicrobials were trimethoprim/sulphamethoxazole in Uganda, and macrolide, lincosamide and streptogramins in Srilanka [30,31]. The high consumption of these subgroups reflects their broader use and availability in treating prevalent infections, which aligns with previous consumption reports [20] and Ethiopian standard treatment guidelines. In addition, another study on antimicrobial use in low- and middle-income countries similarly found high usage of penicillin's and quinolones [37]. However, the decline in the consumption of macrolides and lincosamides from 6.22% in 2020 to 1.34% in 2022 indicates shifting prescribing practices, likely influenced by initial explorations of azithromycin as a potential treatment for COVID-19. The reduced use could be attributed to a decline in COVID-19 cases and a corrective trend as more evidence emerged regarding the low efficacy of azithromycin in treating COVID-19 [38–40].

The increased consumption of extended-spectrum penicillin's (J01CA) and third-generation cephalosporin's (J01DD) observed in Ethiopia during the study period is concerning due to their high potential for AMR. Conversely, resistance to these antimicrobial agents is significantly increasing in Ethiopia. For example, for the most widely consumed parenteral 3$^{rd}$ generation cephalosporin, ceftriaxone, resistant k. pneumonia is > 80%; and extended spectrum beta lactamase producing bacteria's accounted > 30%. These trends underscore the urgent need for antimicrobial stewardship programs that emphasize appropriate prescribing practices, enhanced diagnostic capabilities, and monitoring of resistance patterns [8,9].

The consumption of antimalarial drugs in Ethiopia increased in 2021, rising to 0.136 DID from 0.015 DID in 2020. This increase may be attributed to a resurgence of malaria cases in various regions of the country towards the end of 2021, as indicated by the WHO report, which was followed by a swift national and global response. The decline in consumption in 2022 may be attributed to adequate stock received at the end of 2021, which was recorded as part of the 2021 consumption due to the nature of our data source. Additionally, the seasonal pattern of malaria in Ethiopia, with peaks from September to December after the primary rainy season and from April to May following the secondary rainy season, may have contributed [41].

Oral formulations were predominant, consistently accounting for over 87% each year. The proportion of orally administered antimicrobials increased from 87.96% in 2020 to 91.24% in 2022. This trend toward high consumption of oral formulations might be driven by their convenience, ease of use, and patient compliance. The data highlight the important role of antimicrobial stewardship and patient education in ensuring the appropriate and responsible use of these essential medications, particularly in community settings where regulation and monitoring may be less stringent [42].

The majority of antimicrobials consumed in Ethiopia fell under the Access category of AWaRe classification (> 70% and >66% as per Ethiopian and WHO AWaRe classification, respectively), exceeding the WHO target of >60% AMC in the Access group. The Watch category, which includes antimicrobials with higher resistance potential, constituted around 24–25% of the total consumption. The minimal use of Reserve category antimicrobials, ranging from 0.02% to 0.06%, is encouraging as it indicates limited reliance on last-resort drugs, addressing concerns about AMR. However, the presence of a small proportion of antimicrobials not classified under the Ethiopian AWaRe classification suggests the need for further revision of the AWaRe classification to include all antimicrobials available in the country. Currently, the Ethiopian AWaRe classification only includes antimicrobials listed in the 2020 Essential Medicines List [43]. This need for revision is supported by similar observations in other countries such as Tanzania and Uganda. A higher and almost similar proportion of Access group AMC was reported in the Tanzanian (> 90%) [32] and Ugandan study (> 65%) [31], respectively.

Amoxicillin, ciprofloxacin, metronidazole (used for amaebiasis and anaerobic bacterial infection), procaine benzyl penicillin, and doxycycline accounted for 75% (DU75) of the overall consumption of antimicrobials in Ethiopia during the reporting period except, procaine benzyl penicillin was not in the DU75 for the year 2022. Additional antimicrobials such as azithromycin, cloxacillin, trimethoprim/sulphamethoxazole, and ceftriaxone contributed to 90% (DU90) of overall antimicrobial consumption. The dominant consumption of oral amoxicillin, ciprofloxacin, and metronidazole, and parenteral procaine benzyl penicillin followed by ceftriaxone, underscores the importance of focusing stewardship efforts to ensure their continued effectiveness in Ethiopia. In contrast, the Tanzanian national AMC survey showed a slightly different pattern, with doxycycline followed by amoxicillin, trimethoprim/sulphamethoxazole, erythromycin, and metronidazole making up their DU75, and procaine benzyl penicillin being the leading parenterally administered antimicrobial [32]. Additionally, the previous AMC survey from 2017–2019 in Ethiopia indicated higher consumption of doxycycline followed by norfloxacin, azithromycin and ciprofloxacin [20]. The higher overall three-year consumption of doxycycline in the previous report might be attributed to an acute watery diarrhea outbreak in 2017.

## Strength and limitations of the study

The use of import data and local manufacturing data, sourced from both paper-based and electronic database records using the WHO GLASS AMC survey standardized methodology, use of comprehensive surveillance data from 2020 to 2022 and robust analysis techniques, to estimate national-level consumption was a strength of this study. This approach permits comparisons over time and across countries. However, these strengths must be balanced with a critical understanding of the study's limitations to ensure that the conclusions drawn are both accurate and actionable. Key limitations that may affect the generalizability of the study finding include:

- Antimicrobials without ATC codes and DDD values as per WHO methodology were not included in the survey.

- Imported and locally manufactured data at the national level were used as proxy indicators for use, which may not reflect the actual use of antimicrobials.

- Only antimicrobials imported through recognized routes were included in the study, but there may be antimicrobials entering the country through illegal and unregulated routes, which could affect the real consumption data.

- Some antimicrobials may have been informally exported (i.e., without official records) or expired before use. As a result, not all antimicrobials included in the total supply were necessarily consumed in Ethiopia.

While the study's strengths highlight its valuable contributions, acknowledging its limitations is crucial for interpreting the results and drawing meaningful conclusions. These limitations provide important insights for future research efforts, particularly emphasizing the need for robust data collection, diverse data sources, and a nuanced understanding of the local context.

## Conclusion

The findings from this study have significant implications for AMR prevention and containment strategies in Ethiopia. The high reliance on imported antimicrobials underscores the urgent need for strategies to boost local production, thereby ensuring a stable and sustainable supply of essential medicines. This is critical for supporting the fourth strategic objective of Ethiopia's National Action Plan on AMR, which focuses on the prudent use of antimicrobials. Targeted interventions should be developed to address the consumption patterns observed in different antibacterial subgroups and routes of administration, promoting rational use and reducing the risk of resistance. The dominance of Access category antimicrobials suggests that stewardship efforts should prioritize ensuring their appropriate use while closely monitoring and controlling the use of Watch and Reserve categories to mitigate the development of resistance. These findings should inform policy revisions and the implementation of more effective stewardship programs, both at the national and sub-national levels. Furthermore, ongoing research and surveillance will be essential in adapting these strategies to evolving trends in AMC and AMR.

## Supportive information.

**S1. File. List of antimicrobials WHO ATC/DDD index that lacks ATC code and DDD.** (DOCX)

## Acknowledgments

We would like to thank all staff members at the ports of entry of EFDA for their vital contributions in collecting import data. Their dedication and professionalism were crucial to ensuring the accuracy and reliability of the information gathered. We also thank the local manufacturers and importers for their collaboration and commitment in providing comprehensive antimicrobial data from 2020 to 2022, which was essential for the success of this report. Our sincere appreciation goes to the World Health Organization (WHO) for their invaluable support and guidance throughout the surveillance process. Their contributions were instrumental in ensuring the completion of this report.

## Author contributions

**Conceptualization:** Hailemariam Eshete, Melaku Tileku, Abiyot Aschenaki, Haregewoin Mulugeta, Mengistab Teferi, Teshita Shute, Asnakech Alemu, Heran Gerba, Atalay Mulu Fentie.

**Data curation:** Hailemariam Eshete, Melaku Tileku, Abiyot Aschenaki, Haregewoin Mulugeta, Mengistab Teferi, Teshita Shute, Asnakech Alemu, Heran Gerba.

**Formal analysis:** Hailemariam Eshete, Eshetu Shiferaw, Melaku Tileku, Mengistab Teferi, Atalay Mulu Fentie.

**Funding acquisition:** Heran Gerba.

**Investigation:** Hailemariam Eshete, Abiyot Aschenaki, Haregewoin Mulugeta, Mengistab Teferi, Asnakech Alemu.

**Methodology:** Hailemariam Eshete, Eshetu Shiferaw, Melaku Tileku, Abiyot Aschenaki, Haregewoin Mulugeta, Mengistab Teferi, Asnakech Alemu, Heran Gerba, Atalay Mulu Fentie.

**Project administration:** Abiyot Aschenaki, Haregewoin Mulugeta, Teshita Shute, Asnakech Alemu, Heran Gerba, Atalay Mulu Fentie.

**Resources:** Teshita Shute, Asnakech Alemu, Heran Gerba.

**Supervision:** Eshetu Shiferaw, Melaku Tileku, Mengistab Teferi, Teshita Shute, Asnakech Alemu, Heran Gerba, Atalay Mulu Fentie.

**Validation:** Hailemariam Eshete, Eshetu Shiferaw, Melaku Tileku, Abiyot Aschenaki, Haregewoin Mulugeta, Mengistab Teferi, Teshita Shute, Asnakech Alemu, Heran Gerba, Atalay Mulu Fentie.

**Visualization:** Hailemariam Eshete, Eshetu Shiferaw, Melaku Tileku, Abiyot Aschenaki, Haregewoin Mulugeta, Mengistab Teferi, Teshita Shute, Asnakech Alemu, Heran Gerba, Atalay Mulu Fentie.

**Writing – original draft:** Hailemariam Eshete, Eshetu Shiferaw, Melaku Tileku, Abiyot Aschenaki, Haregewoin Mulugeta, Mengistab Teferi, Teshita Shute, Atalay Mulu Fentie.

**Writing – review & editing:** Hailemariam Eshete, Eshetu Shiferaw, Melaku Tileku, Abiyot Aschenaki, Haregewoin Mulugeta, Mengistab Teferi, Teshita Shute, Asnakech Alemu, Heran Gerba, Atalay Mulu Fentie.

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
