## [Decision Letter · Decision Letter 0]

1 Nov 2024

PONE-D-24-39610Antimicrobial Consumption Trend in Ethiopia: A Surveillance Report 2020-2022PLOS ONE

Dear Dr. Fentie,

Thank you for submitting your manuscript to PLOS ONE. After careful consideration, we feel that it has merit but does not fully meet PLOS ONE’s publication criteria as it currently stands. Therefore, we invite you to submit a revised version of the manuscript that addresses the points raised during the review process.

The reviewer points out that the authors assumed all imported and locally manufactured antimicrobials were consumed domestically, without considering potential exceptions. Specifically:

Exports and Expiration: They suggest that some antimicrobials might have been exported informally (i.e., unrecorded exports) or expired before use. These factors would mean that not all antimicrobials counted in the total supply were actually consumed in Ethiopia.

Recommendation for Study Limitations: The reviewer advises that these factors be acknowledged in the study’s limitations. By adding this to the limitations section, the authors can clarify that the estimated total AMC may not perfectly reflect actual consumption due to these unaccounted variables.

We look forward to receiving your revised manuscript.

Kind regards,

Muhammad Shahzad Aslam, Ph.D.,M.Phil., Pharm-D

Academic Editor

PLOS ONE

2.  Please ensure that you refer to Figure xxxxx in your text as, if accepted, production will need this reference to link the reader to the figure.

Additional Editor Comments:

The reviewer points out that the authors assumed all imported and locally manufactured antimicrobials were consumed domestically, without considering potential exceptions. Specifically:

Exports and Expiration: They suggest that some antimicrobials might have been exported informally (i.e., unrecorded exports) or expired before use. These factors would mean that not all antimicrobials counted in the total supply were actually consumed in Ethiopia.

Recommendation for Study Limitations: The reviewer advises that these factors be acknowledged in the study’s limitations. By adding this to the limitations section, the authors can clarify that the estimated total AMC may not perfectly reflect actual consumption due to these unaccounted variables.

Reviewers' comments:

Reviewer's Responses to Questions

**Comments to the Author**

1. Is the manuscript technically sound, and do the data support the conclusions?

Reviewer #1: Yes

Reviewer #2: Yes

2. Has the statistical analysis been performed appropriately and rigorously? 

Reviewer #1: Yes

Reviewer #2: Yes

3. Have the authors made all data underlying the findings in their manuscript fully available?

Reviewer #1: Yes

Reviewer #2: Yes

4. Is the manuscript presented in an intelligible fashion and written in standard English?

Reviewer #1: Yes

Reviewer #2: Yes

5. Review Comments to the Author

Reviewer #1: Tha authors had done a very good job. I am impressed in how all the data were captured and analysed. The study is novel and is very relevant in the study of AMC and AMR and stewardship especiatlly in the LMIC

Reviewer #2: The work is insightful and the article reported what was on ground for the period of the surveillance. The authors assumed that total quantity of Antimicrobials of interest imported and all manufactured locally equals the consumed Antimicrobials within the period. They didn't make exceptions of those that could have been exported informally and those expired. This estimate quantity should be factored in determining the total AMC. In that regard authors should include it under study limitations.

6. PLOS authors have the option to publish the peer review history of their article (what does this mean? ). If published, this will include your full peer review and any attached files.

**Do you want your identity to be public for this peer review?** For information about this choice, including consent withdrawal, please see our Privacy Policy .

Reviewer #1: **Yes: ** Yejide Olukemi Oseni

Reviewer #2: No

---

## [Author Response · Author response to Decision Letter 0]

2 Nov 2024

Date: November 2, 2024

Point by point response

Dear Editor and Reviewers

We are very grateful for the comments provided for our article entitled “Antimicrobial consumption trend in Ethiopia: A surveillance report 2020-2022” this manuscript.

We revised the manuscript as per the valuable comments and summarized our response to the comments as below along with the an unmarked version of our revised manuscript without tracked changes as “Manuscript” and a marked-up copy of our manuscript that highlights changes made to the original version as “Revised Manuscript with Track Changes”.

Comment Response

Thank you so much. We revised the manuscript as per PLOS ONE requirements (see the marked manuscript copy for your quick reference)

Please ensure that you refer to Figure xxxxx in your text as, if accepted, production will need this reference to link the reader to the figure. Thank you so much and we hereby confirm the figure caption and citation are as per the PLOS ONE author’s guideline.

Please review your reference list to ensure that it is complete and correct. If you have cited papers that have been retracted, please include the rationale for doing so in the manuscript text, or remove these references and replace them with relevant current references. Any changes to the reference list should be mentioned in the rebuttal letter that accompanies your revised manuscript. If you need to cite a retracted article, indicate the article’s retracted status in the References list and also include a citation and full reference for the retraction notice. The reference list is reviewed and hereby confirm the completeness and correctness of the list.

Reviewer #2: The work is insightful and the article reported what was on ground for the period of the surveillance. The authors assumed that total quantity of Antimicrobials of interest imported and all manufactured locally equals the consumed Antimicrobials within the period. They didn't make exceptions of those that could have been exported informally and those expired. This estimate quantity should be factored in determining the total AMC. In that regard authors should include it under study limitations. Indeed thank you so much for the valuable comment. Now, we included the same as a limitation on page 20 line number 324-326 as “Some antimicrobials may have been informally exported (i.e., without official records) or expired before use. As a result, not all antimicrobials included in the total supply were necessarily consumed in Ethiopia.”

---

## [Decision Letter · Decision Letter 1]

16 Dec 2024

PONE-D-24-39610R1Antimicrobial Consumption Trend in Ethiopia: A Surveillance Report 2020-2022PLOS ONE

Dear Dr. Fentie,

Thank you for submitting your manuscript to PLOS ONE. After careful consideration, we feel that it has merit but does not fully meet PLOS ONE’s publication criteria as it currently stands. Therefore, we invite you to submit a revised version of the manuscript that addresses the points raised during the review process.

The peer review comments for the manuscript "Antimicrobial Consumption Trend in Ethiopia: A Surveillance Report 2020-2022" highlighted the need for statistical analysis to strengthen the descriptive nature of the study. Reviewers suggested applying tests like the Mann-Kendall trend test and adding statistical comparisons for 2020, 2021, and 2022 in the tables. The title should use "trends" instead of "trend," and the abstract requires clarification of the "Access group" reference. The introduction should clearly state that the study focuses on human antimicrobial consumption and explain Ethiopia's process for obtaining antimicrobials. The methods section should detail how human-specific data was extracted and clarify Ethiopian AWaRe classification differences from WHO’s classification. Reviewers emphasized the need for transparency regarding antimicrobials lacking ATC codes and suggested tabulating these in the supplementary material.

Results should better define WPP* and clarify units for antimicrobial consumption. Discrepancies in data, like missing D01BA and A07AA, require explanation. The discussion should address AMC trends, including the reasons for increases in 2021 and declines in 2022, with context on how Ethiopia’s experience differs from other countries like Uganda and Tanzania. The increase in usage of extended-spectrum penicillins and third-generation cephalosporins should be linked to potential antimicrobial resistance (AMR) concerns. Reviewers requested more explanation on Ethiopia's antimicrobial supply chain, noting potential community use of antimicrobials during COVID-19.

All drug names should be presented correctly (e.g., "co-trimoxazole" for "sulphamethoxazole + trimethoprim"). Data availability must comply with PLOS Data Policy, ensuring access to raw data or clearly declaring restrictions. The limitations should be clarified to indicate that the focus on human AMC is a methodological choice, not a limitation. Finally, supplementary materials should provide a list of drugs that are not classified under the ATC system, and general grammatical improvements, including merging sentences and ensuring lowercase drug names, were suggested.

We look forward to receiving your revised manuscript.

Kind regards,

Muhammad Shahzad Aslam, Ph.D.,M.Phil., Pharm-D

Academic Editor

PLOS ONE

Additional Editor Comments:

Introduction:

Clearly state that the study focuses on antimicrobial consumption (AMC) in humans.

Provide a brief explanation of Ethiopia’s process for importing and distributing antimicrobials.

Methodology:

Clarify whether the analysis is based solely on human antimicrobial consumption and explain how the data was extracted.

Specify if the "specific data collection points" are selected or all points of entry.

Use "extraction" instead of "abstraction" to avoid confusion.

Provide a citation for the ATC/DDD 2023 index and ensure proper numbering for all references.

Clarify how the Ethiopian AWaRe classification differs from the WHO AWaRe classification and provide a reference.

State how antimicrobials that lack ATC codes and DDDs were handled and consider tabulating this information in a supplementary table.

Address discrepancies in antimicrobial categories and clarify missing data for D01BA (2022) and A07AA (entire study period).

Results:

Clarify the meaning of WPP* with a footnote in Table 1.

Merge two sentences related to “Whereas” (Line 168) into one for clarity.

Provide the rationale for the inclusion of the Ethiopian AWaRe classification and discuss if it is necessary.

Ensure consistency in reporting the units for antimicrobial consumption (e.g., DID, tons, percentage) and clearly label units in tables (e.g., Table 7).

Address missing or excluded antimicrobials from the ATC classification and explain the proportion of total consumption they represent.

Justify the removal of D01BA in 2022 and the absence of A07AA in the study period.

Discussion:

Discuss observed trends in AMC, particularly the increase from 2020 to 2021 and the decrease in 2022.

Explain the rationale for comparing Ethiopian AMC data with Uganda and Tanzania, and clarify how COVID-19 impacted AMC differently in Ethiopia.

Provide possible explanations for the increase in AMC, such as community use of antimicrobials during the COVID-19 pandemic.

Address the significant usage of metronidazole (P01AB) for amoebiasis, noting that it is also used for anaerobic bacterial infections.

Explain why certain antimicrobial agents, like procaine benzyl penicillin, appeared in 2020 and 2021 but not in 2022.

Highlight the implications of increased use of extended-spectrum penicillins (J01CA) and 3rd-generation cephalosporins in the context of antimicrobial resistance (AMR).

Discuss potential reasons for the decrease in consumption of antimalarials (e.g., policy or treatment guideline changes).

Clarify statements about disruptions in Ethiopia's antimicrobial supply during COVID-19 and support these claims with evidence.

Data and Availability:

Ensure that all data supporting the findings are made publicly available or properly declared if restrictions exist.

Clarify data availability and provide a data access link if applicable, in accordance with PLOS Data Policy.

Ensure the raw data used in the analysis is accessible for readers and reviewers.

Supplementary Material:

Include supplementary material to list antimicrobials that are not classified under the ATC system.

Provide a detailed list of drugs that lack ATC codes and DDDs.

Language and Grammar:

Ensure the term "trends" is consistently used instead of "trend" throughout the paper.

Use clear and direct language in the discussion to avoid ambiguous or unsupported claims.

Rephrase "failing to provide a complete picture of antimicrobial consumption" to "these studies did not aim to reveal the complete antimicrobial consumption."

Limitations and Strengths:

Clarify that the focus on human AMC is a methodological decision, not a limitation.

Discuss the generalizability of the results to other countries, especially in terms of resistance patterns and AMC drivers.

Highlight the strengths of the study, such as the use of comprehensive surveillance data from 2020 to 2022.

Reviewers' comments:

Reviewer's Responses to Questions

**Comments to the Author**

1. If the authors have adequately addressed your comments raised in a previous round of review and you feel that this manuscript is now acceptable for publication, you may indicate that here to bypass the “Comments to the Author” section, enter your conflict of interest statement in the “Confidential to Editor” section, and submit your "Accept" recommendation.

Reviewer #3: (No Response)

Reviewer #4: (No Response)

Reviewer #5: All comments have been addressed

Reviewer #6: (No Response)

2. Is the manuscript technically sound, and do the data support the conclusions?

Reviewer #3: Yes

Reviewer #4: Yes

Reviewer #5: Yes

Reviewer #6: Yes

3. Has the statistical analysis been performed appropriately and rigorously? 

Reviewer #3: N/A

Reviewer #4: Yes

Reviewer #5: N/A

Reviewer #6: Yes

4. Have the authors made all data underlying the findings in their manuscript fully available?

Reviewer #3: No

Reviewer #4: No

Reviewer #5: Yes

Reviewer #6: Yes

5. Is the manuscript presented in an intelligible fashion and written in standard English?

Reviewer #3: Yes

Reviewer #4: Yes

Reviewer #5: Yes

Reviewer #6: Yes

6. Review Comments to the Author

Reviewer #3: This study is commendable in that it makes every effort possible to clarify the situation of AMC in Ethiopia based on the information available. On the other hand, since readers in other countries do not have a good understanding of the situation in Ethiopia, further explanation is required to make the results of this study useful information for other countries as well. The main questions are as follows

Information such as the incidence rate of patients with drug-resistant infections and the proportion of drug-resistant bacteria in Ethiopia is not shown. Although the AMC in Ethiopia has been clarified, I would like you to discuss the relationship with drug-resistant bacteria.

I would like you to discuss the reasons for the increase in AMC in Ethiopia. You compare the reported values for Uganda and Tanzania, but readers will want to know the reasons for the change in AMC in Ethiopia. In particular, in many other countries, there are many reports of a decrease in AMC due to the impact of COVID-19. I would like you to discuss why such an impact was not seen. For example, is it possible that ordinary citizens used antibacterial drugs thinking they would also be effective against COVID-19?

Can the general public purchase imported antibacterial drugs without a prescription? The routes by which antibacterial drugs are provided to the general public differ from country to country. Therefore, it would be more helpful if you explained in more detail the routes by which antibacterial drugs are obtained in Ethiopia.

What types of antibacterial drugs are not listed in the ATC classification and not included in this study? What proportion do they make up of the total? They need to be included in this study as supplementary materials or defined independently and included in AMC.

"The PLOS Data policy requires authors to make all data underlying the findings described in their manuscript fully available without restriction, with rare exception (please refer to the Data Availability Statement in the manuscript PDF file)." You have declared that you can use all the data, but you have not provided a link to where the data is stored. If it is difficult to post the raw data, I think you need to declare that correctly.

Reviewer #4: I apologize for the number of suggestions I have made in this review. Pease understand that I was assigned to review this manuscript starting from the revised version (not the original version).

This manuscript describes trends in antimicrobial use in Ethiopia and provides valuable information for combating antimicrobial resistance, as the authors have mentioned. I have highlighted some minor issues to improve the quality of the manuscript.

Page 3, Lines 65―66

Since previous studies did not aim to reveal antimicrobial consumption, I believe the phrase “failing to provide a complete picture of antimicrobial consumption across the country” is not an accurate expression. I recommend that the authors revise it to simply say, “these studies did not aim to reveal the complete antimicrobial consumption across the country.”

Page 4, Lines 92―93

Data regarding drugs lacking ATC codes and DDDs seems important for improving antimicrobial usage. I recommend tabulating these drugs somewhere; the Supplementary Table would be a suitable option.

Table 1

Please explain why D01BA was removed in the 2022 data and why A07AA was missing throughout the study duration.

Table 2

Is it necessary to present antimicrobial volume in tons? Presenting data in tons is useful when human data are combined with animal data but may not be relevant when only human data is presented.

Table 6

We cannot calculate DID using the Ethiopian AWaRe category because the classification was not included in this manuscript. Is the Ethiopian AWaRe category necessary? I believe omitting the data categorized under the Ethiopian AWaRe framework will not negatively impact the manuscript.

Page 6

The authors highlighted trends in antimalarial drug use, described them as “notable” in the text, and provided a figure. However, there is no discussion on antimalarials. Please elaborate on what is “notable” about these changes.

Discussion

Antimicrobial consumption in Ethiopia was quite low, with the DID similar to that of developed countries. As DID in this report was calculated using import and production data, actual consumption may be even lower than the reported value. What reasons do the authors believe account for this difference compared to other African countries?

Reviewer #5: PLOS One review

Thank you for inviting me to review this paper: “Antimicrobial consumption trend in Ethiopia: A surveillance report 2020-2022”

An interesting study which gives the overall trends of AMC in Ethiopia over 3 years. The paper is well written with regards to language and the methods used give scientifically valid observations. However, the discussion can be improved.

My detailed comments are given in the attached document.

Reviewer #6: Thanks for the work done here to provide more information and help our understanding of AMU globally.

My suggestion: Kindly apply statistical analysis to your descriptive tables. Statistically compare data between 2020, 2021, and 2022 or some kind of Mann-Kendall trend test. Add this statistics to a column for each of the Table just before three year average. This will add context to your work, your as currently presented is inherently descriptive.

7. PLOS authors have the option to publish the peer review history of their article (what does this mean? ). If published, this will include your full peer review and any attached files.

**Do you want your identity to be public for this peer review?** For information about this choice, including consent withdrawal, please see our Privacy Policy .

Reviewer #3: No

Reviewer #4: **Yes: ** Yoshiki Kusama

Reviewer #5: **Yes: ** Chandanie Amila Wanigatunge

Reviewer #6: **Yes: ** Babafela Awosile

---

## [Author Response · Author response to Decision Letter 1]

25 Dec 2024

Dear Editor and Reviewers.

A rebuttal point by point response is attached along with the revised manuscript with and without track changes.

Thank you in advance for your time in reviewing our manuscript.

Best regards

---

## [Decision Letter · Decision Letter 2]

14 Jan 2025

PONE-D-24-39610R2Ethiopian antimicrobial consumption trends in human health sector: A surveillance report 2020-2022PLOS ONE

Dear Dr. Fentie,

Thank you for submitting your manuscript to PLOS ONE. After careful consideration, we feel that it has merit but does not fully meet PLOS ONE’s publication criteria as it currently stands. Therefore, we invite you to submit a revised version of the manuscript that addresses the points raised during the review process.

Thank you for your revisions. I appreciate your efforts in addressing the earlier feedback. However, I notice that there is still no hypothesis provided regarding the marked changes in antimalarial drug trends over the study period. In your Discussion section, please include a more detailed analysis of the possible reasons behind these shifts. This expanded discussion will help clarify the factors influencing antimalarial consumption trends and strengthen the overall impact of your findings.

We look forward to receiving your revised manuscript.

Kind regards,

Muhammad Shahzad Aslam, Ph.D.,M.Phil., Pharm-D

Academic Editor

PLOS ONE

Journal Requirements:

Additional Editor Comments:

Thank you for your revisions. I appreciate your efforts in addressing the earlier feedback. However, I notice that there is still no hypothesis provided regarding the marked changes in antimalarial drug trends over the study period. In your Discussion section, please include a more detailed analysis of the possible reasons behind these shifts. This expanded discussion will help clarify the factors influencing antimalarial consumption trends and strengthen the overall impact of your findings.

Reviewers' comments:

Reviewer's Responses to Questions

**Comments to the Author**

1. If the authors have adequately addressed your comments raised in a previous round of review and you feel that this manuscript is now acceptable for publication, you may indicate that here to bypass the “Comments to the Author” section, enter your conflict of interest statement in the “Confidential to Editor” section, and submit your "Accept" recommendation.

Reviewer #3: All comments have been addressed

Reviewer #4: (No Response)

Reviewer #5: All comments have been addressed

Reviewer #6: All comments have been addressed

2. Is the manuscript technically sound, and do the data support the conclusions?

Reviewer #3: Yes

Reviewer #4: Yes

Reviewer #5: Yes

Reviewer #6: Yes

3. Has the statistical analysis been performed appropriately and rigorously? 

Reviewer #3: Yes

Reviewer #4: Yes

Reviewer #5: Yes

Reviewer #6: Yes

4. Have the authors made all data underlying the findings in their manuscript fully available?

Reviewer #3: Yes

Reviewer #4: Yes

Reviewer #5: Yes

Reviewer #6: Yes

5. Is the manuscript presented in an intelligible fashion and written in standard English?

Reviewer #3: Yes

Reviewer #4: Yes

Reviewer #5: Yes

Reviewer #6: Yes

6. Review Comments to the Author

Reviewer #3: (No Response)

Reviewer #4: Thank you to the authors for revising the manuscript based on my suggestions. However, no hypothesis has been provided regarding the trends in antimalarial drugs, which have changed dramatically throughout the study periods. Please include a discussion of the possible reasons for these changes in the Discussion section.

Reviewer #5: Thank you for addressing the comments. The paper reads better now and provides useful insights to AMC of Ethiopia.

Reviewer #6: Thanks for an updated version of the manuscript. However, I will consider the study more descriptive and can serve as a baseline study for future comparison.

7. PLOS authors have the option to publish the peer review history of their article (what does this mean? ). If published, this will include your full peer review and any attached files.

**Do you want your identity to be public for this peer review?** For information about this choice, including consent withdrawal, please see our Privacy Policy .

Reviewer #3: No

Reviewer #4: **Yes: ** Yoshiki Kusama

Reviewer #5: **Yes: ** Chandanie Amila Wanigatunge

Reviewer #6: No

---

## [Decision Letter · Decision Letter 3]

19 Jan 2025

PONE-D-24-39610R3Ethiopian antimicrobial consumption trends in human health sector: A surveillance report 2020-2022PLOS ONE

Dear Dr. Fentie,

Thank you for submitting your manuscript to PLOS ONE. After careful consideration, we feel that it has merit but does not fully meet PLOS ONE’s publication criteria as it currently stands. Therefore, we invite you to submit a revised version of the manuscript that addresses the points raised during the review process. Thank you for your diligent efforts in revising the manuscript and for providing possible explanations regarding the observed increase in antibiotic consumption. However, I would like to clarify that reviewer specifically referred to *antimalarial drugs* , not antibiotics. I kindly request the authors to address the trends in *antimalarial drug use* by offering appropriate explanations and insights.

We look forward to receiving your revised manuscript.

Kind regards,

Muhammad Shahzad Aslam, Ph.D.,M.Phil., Pharm-D

Academic Editor

PLOS ONE

Journal Requirements:

Additional Editor Comments:

I would like the authors to provide explanations for the trends in antimalarial drug use.

Reviewers' comments:

Reviewer's Responses to Questions

**Comments to the Author**

1. If the authors have adequately addressed your comments raised in a previous round of review and you feel that this manuscript is now acceptable for publication, you may indicate that here to bypass the “Comments to the Author” section, enter your conflict of interest statement in the “Confidential to Editor” section, and submit your "Accept" recommendation.

Reviewer #4: (No Response)

2. Is the manuscript technically sound, and do the data support the conclusions?

Reviewer #4: Yes

3. Has the statistical analysis been performed appropriately and rigorously? 

Reviewer #4: Yes

4. Have the authors made all data underlying the findings in their manuscript fully available?

Reviewer #4: Yes

5. Is the manuscript presented in an intelligible fashion and written in standard English?

Reviewer #4: Yes

6. Review Comments to the Author

Reviewer #4: I thank the authors for providing possible explanations for the increase in antibiotic consumption. However, in my previous review, I referred to antimalarial drugs, not antibiotics. I would like the authors to provide explanations for the trends in antimalarial drug use. If the manuscript is revised accordingly, I propose that the editors accept it for publication without requiring further review from me.

7. PLOS authors have the option to publish the peer review history of their article (what does this mean? ). If published, this will include your full peer review and any attached files.

**Do you want your identity to be public for this peer review?** For information about this choice, including consent withdrawal, please see our Privacy Policy .

Reviewer #4: **Yes: ** Yoshiki Kusama

---

## [Author Response · Author response to Decision Letter 3]

28 Jan 2025

Dear Editor and Reviewers

We are very grateful for the comments provided for our article entitled “Antimicrobial consumption trend in Ethiopia: A surveillance report 2020-2022” with manuscript ID: PONE-D-24-39610R3 for the second time.

We revised the manuscript as per the valuable comments and summarized our response to the comments as below along with the an unmarked version of our revised manuscript without tracked changes as “Manuscript” and a marked-up copy of our manuscript that highlights changes made to the original version as “Revised Manuscript with Track Changes”.

Thank you for this valuable comment once again and we are very sorry for the misunderstanding of your query. Now, we tried to include a possible explanation why antimalarial consumption was increased in the year 2021 in Ethiopia (Refer line 310-317).

---

## [Editor Report · Decision Letter 4]

30 Jan 2025

Ethiopian antimicrobial consumption trends in human health sector: A surveillance report 2020-2022

PONE-D-24-39610R4

Dear Dr. Fentie,

We’re pleased to inform you that your manuscript has been judged scientifically suitable for publication and will be formally accepted for publication once it meets all outstanding technical requirements.

Kind regards,

Muhammad Shahzad Aslam, Ph.D.,M.Phil., Pharm-D

Academic Editor

PLOS ONE
---

## [Editor Report · Acceptance letter]

PONE-D-24-39610R4

PLOS ONE

Dear Dr. Fentie,

I'm pleased to inform you that your manuscript has been deemed suitable for publication in PLOS ONE. Congratulations! Your manuscript is now being handed over to our production team.

Kind regards,

on behalf of

Dr. Muhammad Shahzad Aslam

Academic Editor

PLOS ONE